# Interobserver variability in interim PET assessment in Hodgkin lymphoma—reasons and solutions

Thomas W. Georgi[1]☯*, Lars Kurch[1]☯, Dirk Hasenclever[2], Victoria S. Warbey[3], Lucy Pike[3], John Radford[4], Osama Sabri[1], Regine Kluge[1‡], Sally F. Barrington[3‡]

1 Department of Nuclear Medicine, University of Leipzig, Leipzig, Germany, 2 Institute for Medical Informatics, Statistics and Epidemiology, University of Leipzig, Leipzig, Germany, 3 King's College London and Guy's & St Thomas' PET Centre, School of Biomedical Engineering and Imaging Sciences, Kings College London, London, United Kingdom, 4 University of Manchester and Christie National Health Service Foundation Trust, Manchester, United Kingdom

☯ These authors contributed equally to this work.
‡ RK and SFB also contributed equally to this work.
* Thomas.Georgi@medizin.uni-leipzig.de

**Data Availability Statement:** All relevant data are within the paper.

**Funding:** The author(s) received no specific funding for this work.

## Abstract

### Introduction

Interim PET (iPET) assessment is important for response adaptation in Hodgkin lymphoma (HL). The current standard for iPET assessment is the Deauville score (DS). The aim of our study was to evaluate the causes of interobserver variability in assigning the DS for iPET in HL patients and to make suggestions for improvement.

### Methods

All evaluable iPET scans from the RAPID study were re-read by two nuclear physicians, blinded to the results and patient outcomes in the RAPID trial. The iPET scans were assessed visually according to the DS and, thereafter, quantified using the qPET method. All discrepancies of more than one DS level were re-evaluated by both readers to find the reason for the discordant result.

### Results

In 249/441 iPET scans (56%) a concordant visual DS result was achieved. A "minor discrepancy" of one DS level occurred in 144 scans (33%) and a "major discrepancy" of more than one DS level in 48 scans (11%). The main causes for major discrepancies were 1) different interpretation of PET-positive lymph nodes—malignant vs. inflammatory; 2) lesions missed by one reader and 3) different assessment of lesions in activated brown fat tissue. In 51% of the minor discrepancy scans with residual lymphoma uptake, additional quantification resulted in a concordant quantitative DS result.

**Competing interests:** The authors have declared that no competing interests exist.

## Conclusion

Discordant visual DS assessment occurred in 44% of all iPET scans. The main reason for major discrepancies was the different interpretation of PET positive lymph nodes as malignant or inflammatory. Disagreements in evaluation of the hottest residual lymphoma lesion can be solved by the use of semi-quantitative assessment.

## Introduction

Interim positron emission tomography (iPET) with the tracer [18]Fluorine-fluorodeoxyglucose (FDG) has a key role in the response adapted treatment in Hodgkin lymphoma. Pediatric patients with a negative iPET result were not irradiated anymore in the EuroNet-PHL trials [1]. In adult Hodgkin lymphoma patients, the iPET result is used to determine the number of subsequent chemotherapy cycles [2].

The current standard for iPET evaluation is the Lugano classification using the five-point Deauville score (DS) [3, 4]. The DS is based on a visual comparison of the highest residual lymphoma uptake (reference lesion) with that of the mediastinal blood pool and normal liver. Significant interobserver variation has been reported for grading using the visual DS [5, 6]. The additional use of semi-quantitative assessment may increase the interobserver agreement for iPET [7–9].

The aim of this study was to systematically evaluate the causes for interobserver discrepancies in the DS assessment, to determine the influence of semi-quantification on the interobserver concordance rate and to make suggestions to improve the interobserver concordance.

## Methods

### Patients

For this study, hybrid iPET/computed tomography (CT) scans from patients enrolled in the RAPID trial were re-evaluated. The RAPID trial took place in the United Kingdom between 2003 and 2010 and included adult patients with early-stage classical Hodgkin lymphoma [10]. The iPET scans were performed at centres within the UK National Cancer Research Institute PET Network after three cycles of chemotherapy with Adriamycin, Bleomycin, Vinblastine and Dacarbazine (ABVD). Based on the results of central review, iPET-positive RAPID trial patients received a fourth cycle of ABVD and involved field radiotherapy. Interim PET negative patients were randomized 1:1 to receive either radiotherapy or no further treatment. The iPET assessment was done according a five-point scale, which later evolved into the DS [11]. Review of PET data was included in the original protocol, approved as part of the ethics application. All patients enrolled in the RAPID study gave written informed consent. Ethical approval was granted by the North West Multicentre Research Ethics Committee (MREC 03/8/056). For our analysis, all iPET scans were completely anonymized. Thus, identification of patients was not possible.

### Interim PET assessment

All available iPET/CT scans (n = 450) from the RAPID trial patients were re-read independently by two nuclear physicians, blinded to the results and patient outcomes in the RAPID trial. Both nuclear physicians were experienced in interpreting lymphoma scans using the DS with eleven and twelve years expertise, respectively. The iPET/CT scans were assessed visually

and quantitatively using the Hybrid3D viewer software (HERMES Medical Solutions, Stockholm, Sweden).

## Visual assessment

All iPET scans were assessed visually according to the Lugano classification [3].

Baseline PET scans were not performed in the RAPID trial. The initial sites of disease were given on case record forms, populated from the baseline CT scan reports. Sites of initial involvement were recorded by anatomical region as cervical, supraclavicular, axillary, mediastinal, pulmonary hilar and other regions (e.g. inguinal). The visual DS (vDS) was determined using the lymphoma lesion with the highest uptake visually to assign the overall score.

## Quantitative assessment

After the visual assessment, semi-quantitative assessments were performed. All lymphoma lesions with residual uptake (vDS 2–5) were measured by the qPET method [12]. The qPET value is defined as quotient of the average standard uptake value (SUV) of the hottest connected voxels within the lymphoma residual and the mean SUV of the liver using a standard volume of interest. The highest qPET value was used to determine the quantitative DS (qDS) according to the published thresholds [13] (Table 1).

## Interobserver discrepancies

The results of the visual assessments from both readers were compared. Discordant results were discussed between the two readers (LK, TG) and three further nuclear physicians (RK, SB, VW) with long-year experience in the iPET assessment in lymphoma patients in order to find causes for interobserver discrepancies and suggestions to prevent them.

The discordant iPET results were divided into "minor discrepancies" of one vDS level and "major discrepancies" of greater than one vDS level. All iPET scans with a major discrepancy were re-evaluated by both readers sitting side-by-side. In iPET scans with a minor discrepancy, the qDS results were compared.

# Results

## Interim PET assessment

Four hundred and fifty iPET/CT scans were available for re-evaluation. Four hundred forty-one scans were assessed visually and quantitatively by both readers. In five of the remaining nine iPET scans, at least one reader considered that the scan was not assessable due to prominent brown fat activity. In one patient, the image quality of the scan was considered to be inadequate for response assessment by one reader. In the other three patients, quantification was not possible because the data from an older Philips PET scanner were not stored in units of activity concentration and the Hybrid3D program was unable to display data as SUV values.

**Table 1. Classification of the quantitative Deauville score (qDS) according to the qPET value.**

| qDS | qPET value of lesion with highest residual uptake |
|-----|---------------------------------------------------|
| 1   | 0                                                 |
| 2   | $< 0.95$                                           |
| 3   | $\geq 0.95$ and $< 1.30$                           |
| 4   | $\geq 1.30$ and $< 2.00$                           |
| 5   | $\geq 2.00$                                        |

## Comparison of visual assessment

A concordant vDS result was achieved in 249/441 (56%) iPET scans. However, a minor discrepancy occurred in 144 scans (33%) and a major discrepancy in 48 scans (11%) (Table 2). Concerning the binary decision between complete metabolic response (CMR) (vDS 1–3) and non-CMR (vDS 4–5), a concordant result was obtained in 401/441 scans (91%).

## Reasons for discrepancy

The discussion between the nuclear physicians suggested the following causes for visual discrepancies.

**1) Detection-based discrepancies.**

1) 1) Lesion not evaluable

One reader interpreted uptake as residual lymphoma, whilst the other reader considered that this region was not evaluable, e.g. due to the presence of activated brown fat tissue or motion artifacts.

1) 2) Lesion overlooked

One of the two readers interpreted a lesion as residual lymphoma and the other reader overlooked the lesion.

**2) Interpretation-based discrepancies.**

2) 1) Malignant vs. non-pathological

One reader interpreted uptake as residual lymphoma, whilst the other reader interpreted the uptake as physiological or artifactual uptake.

2) 2) Malignant vs. inflammatory

One reader interpreted a lesion as residual lymphoma, the other reader as inflammation or infection.

**3) Evaluation-based discrepancies.**

3) 1) Different hottest lesions

Both readers chose different 'hottest' lesions.

3) 2) Different vDS

The two readers agreed on the location of the 'hottest' residual lesion, yet disagreed in their interpretation of the vDS.

**Table 2. Comparison of the visual Deauville scores (vDS) of reader 1 (R1) and reader 2 (R2).**

|  | R1 vDS 1 | R1 vDS 2 | R1 vDS 3 | R1 vDS 4 | R1 vDS 5 | Sum |
|---|---|---|---|---|---|---|
| R2 vDS 1 | 109 | 32 | 12 | 12 | 0 | 165 |
| R2 vDS 2 | 44 | 81 | 10 | 3 | 0 | 138 |
| R2 vDS 3 | 11 | 33 | 39 | 7 | 2 | 92 |
| R2 vDS 4 | 3 | 3 | 8 | 7 | 8 | 29 |
| R2 vDS 5 | 0 | 1 | 1 | 2 | 13 | 17 |
| Sum | 167 | 150 | 70 | 31 | 23 | 441 |

## Major discrepancies

Overall, 51 major discrepancies occurred in 48 patients. Nineteen major discrepancies were detection-based; eight times "lesion not evaluable" and eleven times "lesion overlooked". Activated brown fat tissue was always the reason for non-evaluable lesions in our patients. Thirty two major discrepancies were interpretation-based; 25 times "malignant vs. inflammatory" and seven times "malignant vs. non-pathological". Non-pathological uptake was observed in the thymus, in cervical muscles and in vessel walls or was artifactual. A concordant qDS was not achieved in any of the patients with a major discrepancy (S1 Table).

## Minor discrepancies

One hundred and forty-four minor discrepancies occurred in our study; 76 discrepancies between vDS 1/2, 43 between vDS 2/3, 15 between vDS 3/4 and ten between vDS 4/5, respectively. In 68 scans with a minor discrepancy, both readers measured at least one qPET-value and the qDS could be compared. In 35 of these 68 scans (51%), a concordant qDS was achieved (Table 3).

## Discussion

The interobserver agreement for the ordinal visual assessment of DS in lymphoma patients is mediocre, concordant results for readers assigning a vDS of 1–5 were reported of only 42% and 62% [5, 6]. In our study, the concordance rate for ordinal vDS was similar at 56%.

The discussion between the two readers (LK,TG) and three further nuclear physicians with extensive experience in the iPET assessment of lymphoma patients (RK, VW, SB) suggested the following causes for visual discrepancies: detection-, interpretation- and evaluation-based causes.

Detection-based discrepancies occur if a lesion is overlooked or is considered to be non-evaluable by one of two or more readers. Around 40% of major discrepancies in our study fell into this category. Reducing the chances of lymphoma lesions being overlooked is challenging but include approaches such as careful comparison of the iPET with baseline PET scan findings [3], review by nuclear physicians experienced in lymphoma reading [14] and the additional use of automatic segmentation algorithms [15]. However, in our opinion, cases where a lesion is truly "overlooked" are rare. It is more likely that the reader actually "sees" the lesion but interprets it subconsciously as non-malignant uptake and passes it over. A second reason for detection-based discrepancies is increased brown fat activity around the residual lymphoma lesion, which can be reduced by administration of a non-selective beta-blocker [16]. Use of warming pads and blankets, hot unsweetened tea and warm intravenous infusions have also been suggested to decrease brown fat activity [17]. Image quality may also be degraded by patient motion, breathing artifacts, incorrect attenuation correction, cutaneous tracer contamination, uptake in intravenous catheters, high blood glucose levels or obese body habitus,

**Table 3. Comparison of the quantitative Deauville score (qDS) in minor visual discrepancies.**

| Minor visual discrepancies | | | |
|---|---|---|---|
| Discrepancy between | N | Concordant qDS | Ratio |
| vDS 1/2 | 76 | - | - |
| vDS 2/3 | 43 | 22 | 51% |
| vDS 3/4 | 15 | 6 | 40% |
| vDS 4/5 | 10 | 7 | 70% |
| Overall (without DS 1/2) | 68 | 35 | 51% |

leading to non-evaluable scans. The influence of some of these factors on image quality can be mitigated by good patient preparation and communication, paying close attention to patient comfort and complying with international guidance for FDG tumor imaging [18].

Interpretation-based discrepancies occur where both readers detect the same lesion yet disagree about lesion aetiology. The differentiation between malignant and inflammatory uptake was the main reason for major discrepancies in our study accounting for 52% of cases. Access to clinical history such as intercurrent infection, trauma, recent vaccinations and access to correlative imaging was missing in our study as readers were blinded to patient information. This may have accounted for the high discrepancy rate in this category. A useful approach for iPET assessment in our opinion is to consider tracer uptake above liver in initially involved lymph nodes or extranodal sites as still malignant, but consider carefully whether new uptake in sites that were not initially involved might have an alternative aetiology such as inflammation or infection, especially if the lymphoma has responded well to therapy overall [19]. This approach relies on having a baseline PET scan. The differentiation between malignant and physiological uptake is also difficult, especially in the thymus, in the Waldeyer´s ring and uptake that lies close to salivary glands, cervical muscles or vessel walls [5]. The corresponding morphological imaging, good coregistration between PET and CT and awareness of the variety of physiological FDG uptake patterns can improve the differentiation between physiological and malignant uptake. Correlative anatomical imaging may also be helpful to distinguish between malignant and artifactual uptake. Increased tracer uptake without a morphological correlate should not be interpreted as a lymphoma lesion.

Evaluation-based discrepancies occur where different visual Deauville scores are assigned because the readers select different lesions for the hottest lesion or they choose the same lesion but score it differently due to a difference in visual perception usually when lesions have similar intensity [12, 20]. Additional quantification should solve evaluation-based discrepancies since the quantification of a lymphomatous lesion yields the same value regardless of the reader, provided the same method for measurement is used and the same lesion is measured. The detection of the hottest lesion is also unambiguous with additional quantification. Provided the reader measures several candidate lesions, the quantified values will determine the hottest lesion and eliminate the visual discrepancy in interpretation of uptake. However, semi-quantitation is unable to solve discrepancies between vDS 1 and 2 since DS 1 means there is no residual uptake measurable. In our study, a concordant qDS result was achieved in 35 of all 192 (18%) discrepancies.

A summary of our practical suggestions for the improvement of the interobserver concordance is given in Table 4.

The iPET result is important for the determination of further treatment [1, 2]. Thus, interobserver discrepancies can have significant clinical consequences. On the one hand, undertreatment of patients increases the risk of recurrence [21]. On the other hand, an increased treatment intensity that is not indicated unjustifiably increases the risk of therapy-associated late effects, especially in children [22]. In our study, the concordance rate for the clinically important decision threshold for categories of CMR (vDS 1–3) versus non-CMR (vDS 4/5) was high at 91%. This is in keeping with published concordance rates for categorical DS assessment of CMR versus non-CMR between 86% and 95% [5, 23–26]. However, in 40 (21%) of all 192 discrepancies in our study, both readers disagreed as to whether the patient achieved a CMR or not. This would have influenced the further treatment. Six of these 40 discrepancies were resolved by additional quantification. Some of the other discrepancies might be also solved by the suggested improvements for detection- and interpretation-based discrepancies.

**Table 4. Causes for interobserver discrepancies in the visual Deauville scoring and suggestions for improvement.**

| | Causes for visual discrepancies | | Suggestion for improvement |
|---|---|---|---|
| Detection-based | Lesion not evaluable, e.g. due to | | |
| | | • activated brown fat tissue | • non-selective beta-blocker |
| | | • patient motion, breathing artifacts | • warming pads and blankets |
| | | | • hot unsweetened tea |
| | | | • warm intravenous infusions |
| | | | • good patient preparation and communication |
| | | • incorrect attenuation correction,high blood glucose levels | • paying close attention to patient comfort |
| | | | • complying with international guidance for FDG tumor imaging |
| | Lesion overlooked | | • comparison with baseline PET scan |
| | | | • review by experienced nuclear physicians |
| | | | • use of automatic segmentation algorithms |
| Interpretation-based | Malignant vs. physiological | | • use of correlative morphological imaging |
| | | | • good coregistration between PET and CT |
| | | | • awareness of the variety of physiological FDG uptake patterns |
| | Malignant vs. inflammation | | • access to clinical history (intercurrent infection, trauma, recent vaccinations) |
| | | | • use of correlative morphological imaging |
| | | | • comparison with baseline PET scan |
| | | | • definition of clear assessment algorithms |
| Evaluation-based | Different hottest lesions | | • additional use of a quantitative method |
| | Different visual DS of the same lesion | | |

## Limitations

A major limitation was the lack of baseline PET scans. A comparison with the baseline PET scan would likely have decreased the number of overlooked lesions and improved the differentiation between malignant and non-malignant uptake by mapping the areas on the response scan to those with high-grade lymphomatous involvement pre-treatment.

Both readers in this study were from the same department of nuclear medicine with similar training in iPET assessment in lymphoma patients. The discrepancy rate might be higher for readers from different institutions.

The RAPID trial enrolled early-stage Hodgkin lymphoma patients and hence our study was based only on iPET assessment of nodal lesions since patients with extranodal involvement were ineligible.

## Conclusion

Discordant visual assessment between the two readers in the ordinal DS occurred in 44% of all iPET scans. Causes for visual discrepancy can be detection-, interpretation- and evaluation-based.

Detection- (lesion not evaluable, lesion overlooked) and interpretation-based discrepancies (malignant vs. inflammatory/physiological) are unaffected by semi-quantitative evaluation but may be reduced by comparing response with baseline scans, improved reader training, review of morphological imaging, knowledge of clinical history and close attention to patient preparation to reduce factors that can affect image quality, especially brown fat.

Evaluation-based discrepancies (different hottest lesions, different visual DS) can be solved by a semi-quantitative assessment, provided the readers measured the same lesions. Our

suggestion would be to use a quantitative method additionally to the visual assessment to decrease evaluation-based discrepancies.

## Supporting information

**S1 Table. Comparison of the quantitative Deauville score (qDS) in major visual discrepancies.**
(PDF)

## Author Contributions

**Conceptualization:** Thomas W. Georgi, Dirk Hasenclever, Victoria S. Warbey, Lucy Pike, John Radford, Regine Kluge, Sally F. Barrington.

**Data curation:** Thomas W. Georgi, Lars Kurch.

**Formal analysis:** Thomas W. Georgi, Dirk Hasenclever.

**Investigation:** Thomas W. Georgi, Lars Kurch.

**Methodology:** Thomas W. Georgi.

**Project administration:** John Radford, Osama Sabri, Regine Kluge, Sally F. Barrington.

**Supervision:** Dirk Hasenclever, John Radford, Osama Sabri, Regine Kluge, Sally F. Barrington.

**Validation:** Thomas W. Georgi, Lars Kurch, Dirk Hasenclever, Victoria S. Warbey, Lucy Pike, Sally F. Barrington.

**Visualization:** Thomas W. Georgi.

**Writing – original draft:** Thomas W. Georgi.

**Writing – review & editing:** Lars Kurch, Dirk Hasenclever, Victoria S. Warbey, Lucy Pike, John Radford, Osama Sabri, Regine Kluge, Sally F. Barrington.

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
