## [Decision Letter · Decision Letter 0]

8 Jan 2023

PONE-D-22-33844Interobserver variability in interim PET assessment in Hodgkin lymphoma - Reasons and solutionsPLOS ONE

Dear Dr. Georgi,

Thank you for submitting your manuscript to PLOS ONE. After careful consideration, we feel that it has merit but does not fully meet PLOS ONE’s publication criteria as it currently stands. Therefore, we invite you to submit a revised version of the manuscript that addresses the points raised during the review process.

We look forward to receiving your revised manuscript.

Kind regards,

Domenico Albano

Academic Editor

PLOS ONE

Reviewers' comments:

Reviewer's Responses to Questions

**Comments to the Author**

1. Is the manuscript technically sound, and do the data support the conclusions?

Reviewer #1: Partly

Reviewer #2: Partly

2. Has the statistical analysis been performed appropriately and rigorously? 

Reviewer #1: Yes

Reviewer #2: N/A

3. Have the authors made all data underlying the findings in their manuscript fully available?

Reviewer #1: Yes

Reviewer #2: Yes

4. Is the manuscript presented in an intelligible fashion and written in standard English?

Reviewer #1: Yes

Reviewer #2: Yes

5. Review Comments to the Author

Reviewer #1: Dear authors,

this is a really interesting work focusing on an interest issue of nuclear medicine as you underline in the introduction and in the discussion.

The english language and the syntax are good.

I have some minor issues to underline:

- why not perform the qDS also in the case of major discrepancies?

- in the conclusione you stated that semiquantitative assessment can solve disagreement, however in your study this is true only in a small amount of the cases; please correct.

- a minor point is that the acronym SUV is used without a definition.

Reviewer #2: This manuscript describes the results of a blinded iPET scan review conducted by two experienced nuclear physician of scans collected from the Rapid randomized trial for early stage HL. Authors describe the results of the two readers and report on the reasons for major and minor discrepancies. They also describe adoption of qPET as a meaningful method to solve some of the discrepancies. In the discussion they provide some suggestion to reduce discrepancies.

Overall this is simple descriptive study that allows to define some practical suggestion that, however, would require validation or at least an expert consensus. From a clinical point of view this study adds little to what is currently known and does not really help to identify a final key message. Authors should try to answer the main question that for a clinician is...what is the clinical consequence of observed discrepancies? Which are pactical suggestions to move the field forward? Are suggestions able to reduce the effect of discrepancies for the final clinical decision?

One practical suggestion is to add a table with a summary of practical suggestion identified to reduce discrepancies.

One relevant aspect here is that baseline PET scans were missing. How do authors believe availability of baseline scan would have helped to reduce discrepancies?

The main conclusion with this paper is about the utility of qPET to solve some discrepancies. To be pragmatic does this mean that qPET should be used from the beginning instead of the conventional visual approach recommended for the DS definition?

6. PLOS authors have the option to publish the peer review history of their article (what does this mean?). If published, this will include your full peer review and any attached files.

Reviewer #1: No

Reviewer #2: No

---

## [Author Response · Author response to Decision Letter 0]

15 Feb 2023

Point-by-point reply to the reviewers’ comments

Reviewer #1: 

Why not perform the qDS also in the case of major discrepancies?

We performed the qDS also in the patients with a major discrepancy. A concordant qDS was not achieved in any of the patients with a major discrepancy. We added this information in the result part of the manuscript (line 170-171) and in a table (S1 Table) in the supplementary material.

In the conclusion you stated that semiquantitative assessment can solve disagreement, however in your study this is true only in a small amount of the cases; please correct.

You are right, only a minor number of discrepancies 35/192 (18%) can be solved by semiquantitative assessment.

The majority of discrepancies are detection- or interpretation-based and can not be solved by additional quantification since lesions overlooked or not considered to be lymphoma are not quantified. This fact is given in the second paragraph of the conclusion (line 267-268). 

However, we think that our statement in the last paragraph of the conclusion is correct (line 272-273) and evaluation-based discrepancies can be solved by additional quantification, if both readers measured the same lesion. Evaluation-based discrepancies occur if both readers agree on the lymphoma residuals, yet disagree which of them is the ´hottest´ lesion or which is the visual DS of the hottest lesion. The use of a quantitative method provides a specific value for each lesion. These values determine the order of the lesions and, thus, the ´hottest´ lesion regardless of the reader. Moreover, quantified lesions can be classified into a qDS according to validated cut-off values.

A minor point is that the acronym SUV is used without a definition.

The acronym SUV is written out now (line 100). Thank you for this remark.

Reviewer #2: 

Practical suggestion, however, would require validation or at least an expert consensus. 

The discordant iPET results were discussed between the two readers (LK, TG) and three further authors of our manuscript (SB, RK, VW) in order to find causes for interobserver discrepancies. Our group consists of five nuclear physicians with long-year experience in the iPET assessment in lymphoma patients, e.g. reference reading of all PET scans in the EuroNet-PHL-C1 and -C2 trial with pediatric patients with Hodgkin lymphoma (RK, LK, TG) and in the RAPID trial with adult Hodgkin lymphoma patients (SB, VW). Therefore, our practical suggestions can be seen as an expert consensus.

This fact has not yet been clearly stated in the manuscript. We added this information in the method part (line 110-112) and in the discussion part (line 187-188) of the manuscript.

What is the clinical consequence of observed discrepancies? 

The iPET assessment plays an important part in the determination of further lymphoma treatment. Pediatric patients with Hodgkin lymphoma with a negative iPET result were not irradiated anymore in the EuroNet-PHL trials. In adult patients, the iPET result is used to determine the number of subsequent chemotherapy cycles. Thus, interobserver discrepancies can have significant clinical consequences. On the one hand, undertreatment of patients increases the risk of recurrence. On the other hand, an increased treatment intensity that is not indicated unjustifiably increases the risk of therapy-associated late effects, especially in children.

The current threshold between a positive and negative iPET result is between DS 3 and 4. In 40 (21%) of all 192 discrepancies in our study, both readers disagreed as to whether the iPET was positive or negative. This would have led to different further treatment intensities. 

We added these informations in the introduction (line 54-56) and discussion (line 241-244 and 247-249) part of the manuscript.

Are suggestions able to reduce the effect of discrepancies for the final clinical decision?

Yes, our suggestions are able to reduce discrepancies for the final clinical decision. As mentioned above, In 40 patients, both readers disagreed as to whether the iPET was positive or negative. Six of these 40 discrepancies were resolved by additional quantification. Some of the other discrepancies might be also solved by our suggested improvements. We added this information in the discussion part (line 249-251) of the manuscript. 

According to your remark, we added a table with practical suggestions to reduce interobserver discrepancies (table 4). In return, we have dispensed with Figure 1.

How do authors believe availability of baseline scan would have helped to reduce discrepancies?

First, the likelihood of missing a lymphoma lesion in the iPET scan is significantly reduced because the reader is looking for every lesion visible in the baseline PET scan.

Second, the differentiation between malignant and inflammatory tracer uptake is much easier. Increased tracer uptake in initially involved lymph nodes is most probably malignant. Whereas increased uptake in initially not involved lymph nodes is more likely to be inflammatory if the lymphoma is generally responding well to treatment.

Thus, baseline PET scan is suitable to reduce both, detection- and interpretation-based discrepancies. 

Does this mean that qPET should be used from the beginning instead of the conventional visual approach recommended for the DS definition?

No, our suggestion would be the use of a quantitative method additionally to the visual assessment. This would decrease evaluation-based discrepancies. 35/192 (18%) visual discrepancies in our study achieved a concordant qDS.

---

## [Decision Letter · Decision Letter 1]

14 Mar 2023

Interobserver variability in interim PET assessment in Hodgkin lymphoma - Reasons and solutions

PONE-D-22-33844R1

Dear Dr. Thomas W Georgi,

We’re pleased to inform you that your manuscript has been judged scientifically suitable for publication and will be formally accepted for publication once it meets all outstanding technical requirements.

Kind regards,

Domenico Albano

Academic Editor

PLOS ON

Reviewer's Responses to Questions

**Comments to the Author**

1. If the authors have adequately addressed your comments raised in a previous round of review and you feel that this manuscript is now acceptable for publication, you may indicate that here to bypass the “Comments to the Author” section, enter your conflict of interest statement in the “Confidential to Editor” section, and submit your "Accept" recommendation.

Reviewer #1: All comments have been addressed

2. Is the manuscript technically sound, and do the data support the conclusions?

Reviewer #1: Yes

3. Has the statistical analysis been performed appropriately and rigorously? 

Reviewer #1: Yes

4. Have the authors made all data underlying the findings in their manuscript fully available?

Reviewer #1: Yes

5. Is the manuscript presented in an intelligible fashion and written in standard English?

Reviewer #1: Yes

6. Review Comments to the Author

Reviewer #1: (No Response)

7. PLOS authors have the option to publish the peer review history of their article (what does this mean?). If published, this will include your full peer review and any attached files.

Reviewer #1: No

---

## [Editor Report · Acceptance letter]

17 Mar 2023

PONE-D-22-33844R1 

Interobserver variability in interim PET assessment in Hodgkin lymphoma - Reasons and solutions 

Dear Dr. Georgi:

I'm pleased to inform you that your manuscript has been deemed suitable for publication in PLOS ONE. Congratulations! Your manuscript is now with our production department. 

Kind regards, 

on behalf of

Dr. Domenico Albano 

Academic Editor

PLOS ONE